# Verbal and psychological violence against women in Turkey and its determinants

Ömer Alkan[1¤a]*, Ceyhun Serçemeli[2©¤b], Kenan Özmen[3©]

1 Department of Econometrics, Ataturk University, Erzurum, Turkey, 2 Department of Labor Economics and Industrial Relations, Ataturk University, Erzurum, Turkey, 3 Bulanik Vocational School, Mus Alparslan University, Muş, Turkey

© These authors contributed equally to this work.
¤a Current address: Department of Econometrics, Faculty of Economics and Administrative Sciences, Ataturk University, Erzurum, Turkey
¤b Current address: Department of Labor Economics and Industrial Relations, Faculty of Economics and Administrative Sciences, Ataturk University, Erzurum, Turkey
* oalkan@atauni.edu.tr

**Data Availability Statement:** The data underlying this study is subject to third-party restrictions by the Turkey Statistical Institute. Data are available from the Turkish Statistical Institute (bilgi@tuik.

## Abstract

Verbal and psychological violence against women is considered an important sociological and legal problem and a serious threat within the context of basic human rights. The aim of this study was to detect the factors affecting verbal and psychological violence against women in Turkey, a developing country. The micro data set of the National research on domestic violence against women in Turkey, which was conducted by the Hacettepe University Institute of Population Studies, was employed in this study. The factors affecting women's exposure to verbal and psychological violence by their husbands or partners in Turkey were determined using binary logistic and binary probit regression analyses. Women whose husbands or partners cheated and used alcohol were more exposed to verbal and psychological violence compared to others. In addition, women who were exposed to physical, economic, and sexual violence were more exposed to verbal and psychological violence compared to others. Exposure to violence by first-degree relatives increases the possibility of exposure to verbal and psychological violence. More effective results can be achieved by prioritizing women likelier to be exposed to violence in policies aimed at preventing acts of verbal violence against women in our country. There are few studies on verbal and psychological violence against women. Therefore, it will be useful to conduct relevant studies from different perspectives.

## Introduction

Violence against women is considered a highly complicated issue and a multidimensional problem [1]. Even though its form differs from one society and culture to another, violence against women has always persisted [2]. It is evident that violent acts are growing more prevalent in today's social lives [3–6]. Violence is an embarrassing phenomenon that is observed in all areas of society, including the streets, schools, workplaces, and homes, and has become

gov.tr) for researchers who meet the criteria for access to confidential data. The authors of the study did not receive any special privileges in accessing the data.

**Funding:** The authors received no specific funding for this work.

**Competing interests:** There are no conflicts of interest to declare.

universal [7]. Violence against women can be defined as gender-based acts of violence that are used by a man, cause only the woman to suffer, and appear as physical, sexual, verbal, psychological and/or economic symptoms. The perpetrators appear to be partners, neighbors, friends, relatives, colleagues, and foreigners. Nevertheless, the family environment is usually the arena where men use most violence against women and girls [8]. A report by the World Health Organization (WHO) indicated that violence against women was widespread on a global level and caused serious health problems. Considering the results included in the report, it is understood that violence against women is not a minor problem that is observed only in a certain part of society but a global public health problem that requires immediate action. Violence against women has many effects on health, from minor physical injuries to traumas that may result in death. Psychological disorders may also lead to severe effects, such as post-traumatic stress disorder (PTSD), depression, and substance use [9]. Approximately 1.3 million adolescents die from infectious diseases, injuries, pregnancy, and childbirth every year worldwide. Moreover, 45 percent (approximately 600,000) of these deaths are among adolescent girls, and violence causes about 10 percent of these deaths [10].

The Centers for Disease Control and Prevention in the United States define intimate partner violence as physical, sexual, and/or emotional violence, abuse, or threats used by people in close contact, including existing or ex-husbands or extramarital partners [11]. Violence against women constitutes a very important problem for societies, and unfortunately, it is still spreading around the world. Nowadays, this issue has been discussed by diverse disciplines, from the social sciences to the juridical sciences, due to mass media and other types of communication [12]. Although violence is a concept that varies with time and sociocultural structure, it has been one of the most crucial issues in recent years. Although domestic violence against women was not an international issue that attracted attention or caused anxiety until half a century ago, this situation changed due to women's rights groups after the 1980s. Violence against women is a significant public health problem and a serious threat to human rights. The United Nations defines violence against women as any act of gender-based violence that results in, or is likely to result in, physical, sexual, or mental harm or suffering to women, including threats of such acts, coercion, or the arbitrary deprivation of liberty, whether in public or in private life [13]. Aggression has a critical effect at the individual and social levels. In general, studies have concluded that men are likelier to be involved in physical aggression than women. Consequently, studies investigating the risk factors for aggressive behavior have largely focused on male populations [14]. The reduction of violence may augment people's well-being and notably reduce public expenditures [15].

Violence can be practiced not only physically but also verbally and psychologically [16–18]. Verbal and psychological violence harms social lives and may include nonverbal threats (shaking fingers, making annoying signs, etc.) or verbal threats (speaking, shouting, swearing, angrily and in an angry tone, etc.). Studies have indicated that women are more vulnerable to violence than men [19]. Nowadays, many studies have been conducted to prevent verbal violence [20]. Studies on nurses working in a profession dominated by women indicated that the risk of nurses being exposed to violence in the workplace was three times higher compared to other occupational groups, and more than one-tenth of nurses have experienced at least one form of violence. The most frequently observed form of violence was reported to be verbal. It has been stated that verbal violence is commonly observed, especially in terms of emergency and intensive care unit employees [21, 22].

The prevalence of psychological and verbal violence against women varies by country [3–5, 23–33]. Violence against women, a global concern, is also one of Turkey's most pressing societal issues [34]. Most studies on violence against women in Turkey are based on the testimonies of women subjected to violence [35]. Compared to sexual and physical violence, comparatively

less research has examined psychological and verbal violence [36]. Studies on verbal and emotional violence against women in Turkey have yielded varying outcomes at the provincial and local levels [13, 37–49].

Verbal and psychological violence is associated with many factors. The level of education of women or their partners subjected to violence is one of these factors [31, 32]. Verbal and psychological violence is also associated with the ages of women and their partners [32, 50]. The financial condition of women or their families is another component of verbal and psychological violence [26, 51]. Place of residence is one of the factors related to the verbal and psychological violence to which women are exposed [52, 53]. Stress and anxiety are elements closely associated with verbal and psychological violence [4].

In a study examining the prevalence of childhood violence and intimate partner violence among 18–24-year-old adolescent girls and young women in Namibia, those who had experienced any form of childhood violence, including verbal and psychological violence, were statistically significantly likelier to experience violence [54]. Moreover, alcohol and cigarette use is one of the aspects related to the verbal and psychological violence that women face [55, 56]. Women with smoking partners are also likelier to experience psychological or physical violence [57]. In addition, having a large number of children and being polygamous (married to more than one woman) heighten the chances of psychological and verbal violence against women [48]. Another study found that exposure to verbal and psychological violence during pregnancy is strongly associated with depression [58].

The aim of this study was to detect the socio-demographic and economic factors affecting women's exposure to verbal and psychological violence by their husbands or partners in Turkey. Furthermore, this study will also determine the characteristics of women's husbands or partners regarding verbal and psychological violence.

## Methods

### Study design

In 2008, a comprehensive report National research on domestic violence against women in Turkey, took place for the first time to define the dimensions of violence against women, identify its causes, and meet the need for data collection on this issue. National research on domestic violence against women in Turkey, conducted in 2014, is significant in its reflection of changes in violence against women since the 2008 study. National research on domestic violence against women in Turkey is one of the most comprehensive studies to understand the magnitude, content, causes, and consequences of domestic violence experienced by women, as well as the risk factors [59, 60].

The research questionnaire was designed by considering the questionnaires used by WHO's Multi-country study on women's healthon Women's Health and domestic violence against women [61]. New questions were added to the questionnaire according to the needs of the country, with a focus on legal compliance [59, 60].

### Setting

Within the scope of the research on violence, Turkey was divided into 30 strata to provide estimates at the national, urban, or rural, 12 regional, and five regional levels. In the research, settlements with a population of 10,000 or more constituted urban strata, and settlements with a population of less than 10,000 were considered rural strata. The research sample consisted of cluster sampling [59, 60].

The field application of the study in 2008 started on July 27, 2008, and was completed on September 29, 2008 [59]. The field application of the study in 2014 started on April 8, 2014, and was completed on July 11, 2014 [60].

## Participants

National research on domestic violence against women in Turkey investigated women between the ages of 15–59. In this study, women who are married, in a relationship, or previously in a relationship were included in the analysis. Women who had never been in a relationship were excluded from the study.

## Data sources/measurement

This present study was a secondary data analysis. This study used the cross-sectional data of National research on domestic violence against women in Turkey, conducted by the Hacettepe University Institute of Population Studies in 2008 and 2014.

In National research on domestic violence against women in Turkey, the research team administered questionnaires in Turkish. The ethical rules developed by WHO were applied at every stage of the research, and various measures were taken to ensure the safety of both the interviewed women and the research team. Before each interview, the consent of respondents was obtained, and the interviewees signed the questionnaire, indicating that this consent was obtained. The researchers were trained in the Code of Ethics and Safety, and were mindful of the subject's sensitivity at the beginning of the interview, during the interview process, and after the interview. If there was more than one woman in the 15–59 age range in the household, a random selection approach was used to avoid asking the same questions to several women, and interviews took place with a single woman from each household. The research teams were quite careful to ensure that the interviews were administered in an environment with only the subjects. All interviewees received training on interview confidentiality. In addition, respondents were notified that their responses would be kept confidential during the approval and dissemination phases [59, 60].

## Study size

In the 2008 study, 12,795 women were interviewed face to face to complete the women's questionnaire, with a rejection rate of 2.1%. The response rate for interviews with women is 86.1% [59]. In the 2014 study, 7,462 women were interviewed face to face, and their questionnaires were filled out, with a rejection rate of 4.4%. The response rate for interviews with women is 83.3% [60]. The computational weights of women were added to these data sets according to the research sample design. Each cluster was assigned a different weight; the reasons for this can be summarized as follows: 1) differential selection probabilities at the cluster level; 2) the non-proportional distribution of the sample size, and 3) differential response rates in each stratum [59, 60].

## Measures and variables

In the National research on domestic violence against women in Turkey, women were asked the following questions: "Did your husband/partner make you sad by swearing at you?", "Did he insult or humiliate you in front of others?", "Did he scare or threaten you? (for instance, by gazing, shouting or breaking things down)?", and "Did he threaten you or your relatives with harm?". The status of exposure to violence measured by these questions was used to generate the dependent variable. The women in the study were exposed to verbal and psychological

violence by their husbands or partners if they experienced at least one of the above-mentioned conditions, and they were not exposed to verbal and psychological violence if they did not experience any of them. In conclusion, the dependent variable of the study was the status of exposure to verbal and psychological violence of the women who received a code 1 if they were exposed to verbal and psychological violence and a code 0 if they were not exposed to it.

The independent variables in this study were detected from variables included in the National research on domestic violence against women in Turkey. The variables related to the socio-demographic and economic characteristics of women were survey year (2008, 2014), region (West, South, Middle, North, East), woman's place of residence (rural, urban), age (15–24, 25–34, 35–44, 45–54, 55+), educational level (illiterate, elementary school, secondary school, high school, university), individual earning and income status (yes, no), health insurance status (yes, no), marital status (never married, once, two and more), number of children owned (has no child, one child, two and more), status of exposure to violence by first-degree relatives (no, yes), and health status (excellent/good, reasonable, bad/very bad).

The factors related to women's husbands or partners were husband or partner's education (illiterate, elementary school, secondary school, high school, university), husband or partner's employment status (no, yes), husband or partner's alcohol use status (no, yes), husband or partner's gambling status (no, yes), husband or partner's drug use status (no, yes), whether husband or partner had cheated (no, yes), status of exposure to husband or partner's economic violence at any point in her life (no, yes), status of exposure to husband or partner's physical violence at any point in her life (no, yes), and status of exposure to husband or partner's sexual violence at any point in her life (no, yes).

## Statistical analysis

Survey statistics in Stata 15 (Stata Corporation) were used to consider the complex sampling design and weights. A weighted analysis was performed [62]. Firstly, the frequency and percentages were obtained according to the status of the exposure to verbal and psychological violence of women participating in the study. Additionally, bivariate analyses determined the relationships between the outcome variable (exposure to verbal and psychological violence) and various factors. We estimated bivariate relationships by evaluating significant differences between categorical variables using Pearson's chi-square test. The Pearson chi-square ($\chi2$) not only gives information regarding the importance of observed distinctions, but also the categories from which any observed differences originate [63].

Subsequently, the risk factors affecting women's exposure to verbal and psychological violence were detected by employing binary logistic and binary probit regression analyses [64]. Binary logit and binary probit models are discrete choice models used when the outcome variable is binary or dichotomous and only takes 0 or 1 [65]. The statistical significance of each independent variable as a risk factor and the ability to calculate the odds ratio were evaluated in binary logistic regression. The cumulative logistic distribution function is used in the binary logit model, and the cumulative normal distribution function (CDF) is used in the probit model. The fact that normal CDF contains integral calculations is cited as a factor leading to a more widespread use of logistic CDF in practice [66].

Ordinal and nominal variables were defined as dummy variables with the aim of observing the effects of the categories belonging to all variables to be taken into logistic and probit regression models [67, 68]. The problem of multicollinearity in the models was considered while identifying the reference category for ordinal and nominal variables with more than two categories. In this regard, the best model was estimated. Therefore, a consistent criterion cannot be selected [69, 70].

Whether there was multicollinearity between the independent variables in the models was also tested. Those with a variance inflation factor (VIF) value of 5 and above were considered to lead to moderate multicollinearity, while those with a value of 10 and above led to high multicollinearity [71].

## Results

### Descriptive statistics and bivariate analysis

The results of the socio-demographic and economic factors that may affect the status of the women's exposure to verbal and psychological violence of the women in Turkey are presented in Table 1. According to the results of the Chi-square test of independence, a significant relationship was found between individuals' exposure to verbal and psychological violence and the socio-demographic and economic variables (except place of residence, individual earning and income) in the study. According to the results of the chi-square test of independence, a significant relationship was found between individuals' exposure to verbal and psychological violence and the factors related to husband or partner in the study.

According to Table 1, while the prevalence of women who participated in the National research on domestic violence against women in Turkey in 2008 was 63.3%, the ratio of those who took part in it in 2014 was 36.7%. Out of 72.1% of women in the study, lived in cities. Most individuals reside in the Western region. The majority of women (78.6%) had no individual earning and income. Most women (85.4) had health insurance. It was detected that 88.8% of women were married only once, and that 69.8% of them had two and more children. While 48.5% of women were elementary school graduates, 9.3% of were university graduates. A total of 11.6% of women were exposed to violence by their first-degree relatives.

It was found that while 72.6% of women who were exposed to verbal and psychological violence by their husbands or partners resided in urban areas, 28.9% of them were from the Eastern Region, 51% of them were elementary school graduates, 78.3% of them had no individual earning and income, 84.1% of them had health insurance, 89.5% of them were married once, 75.6% of them had two and more children, and 16.4% of them were exposed to violence by their first-degree relatives.

While 42.2% of women's husbands or partners were elementary school graduates, 15% of them were university graduates. Table 1 demonstrates that 81.5% of women's husbands or partners were employed, 20.7% of their husbands or partners used alcohol, 2.1% of their husbands or partners gambled, 8.9% of women's husbands or partners cheated, 27.7% of them were exposed to economic violence, 36.7% of them were exposed to physical violence, and 14.1% of them were exposed to sexual violence.

The data proved that the husbands or partners of 45.3% of those who were exposed to verbal and psychological violence by their husbands or partners, were elementary school graduates. The husbands or partners of 80.6% of them were employed, the husbands or partners of 26.4% of them used alcohol, the husbands or partners of 3.8% of them gambled, 15.7% of them had husbands or partners who cheated, 41.2% of them were exposed to economic violence, 65.1% of them were exposed to physical violence by their husbands or partners, and 14.1% of them were exposed to sexual violence by their husbands or partners.

### Estimation of models

Binary logistic regression and binary probit regression models were employed to detect the factors affecting the status of women's exposure to verbal and psychological violence. The estimated model binary logistic and binary probit regression model results are presented in Table 2 and S1 Appendix.

**Table 1. Findings related factors affecting women's exposure to verbal and psychological violence.**

| Variables | | Exposure to verbal and psychological violence | | n (%) | χ2 | P |
|---|---|---|---|---|---|---|
| | | No | Yes | | | |
| **Survey year** | | | | | | |
| | 2008 | 6,528 (62.1) | 5,194 (64.8) | 11,722 (63.3) | 14.178 | < 0.0001 |
| | 2014 | 3,978 (37.9) | 2,818 (35.2) | 6,796 (36.7) | | |
| **Region** | | | | | | |
| | West | 3,213 (30.6) | 2,088 (26.1) | 5,301 (28.6) | 96.945 | < 0.0001 |
| | South | 908 (8.6) | 710 (8.9) | 1618 (8.7) | | |
| | Middle | 2,206 (21.0) | 1,975 (24.7) | 4,181 (22.6) | | |
| | North | 1,471 (14.0) | 925 (11.5) | 2,396 (12.9) | | |
| | East | 2,708 (25.8) | 2,314 (28.9) | 5,022 (27.1) | | |
| **Place of residence** | | | | | | |
| | Urban | 7,538 (71.7) | 5,820 (72.6) | 13,358 (72.1) | 1.798 | 0.180 |
| | Rural | 2,968 (28.3) | 2,192 (27.4) | 5,160 (27.9) | | |
| **Age** | | | | | | |
| | 15–24 | 1,831 (17.4) | 964 (12.0) | 2,795 (15.1) | 118.274 | < 0.0001 |
| | 25–34 | 3,313 (31.5) | 2,544 (31.8) | 5,857 (31.6) | | |
| | 35–44 | 2,726 (25.9) | 2,165 (27.0) | 4,891 (26.4) | | |
| | 45–54 | 1,922 (18.3) | 1,711 (21.4) | 3,633 (19.6) | | |
| | 55+ | 714 (6.8) | 628 (7.8) | 1,342 (7.2) | | |
| **Educational level** | | | | | | |
| | Illiterate | 1,590 (15.1) | 1,424 (17.8) | 3,014 (16.3) | 158.667 | < 0.0001 |
| | Elementary school | 4,905 (46.7) | 4,081 (51.0) | 8,986 (48.5) | | |
| | Secondary school | 1,017 (9.7) | 803 (10.0) | 1,820 (9.8) | | |
| | High school | 1,814 (17.3) | 1,164 (14.5) | 2,978 (16.1) | | |
| | University | 1,180 (11.2) | 537 (6.7) | 1,717 (9.3) | | |
| **Individual earning/income** | | | | | | |
| | No | 8,281 (78.8) | 6,271 (78.3) | 14,552 (78.6) | 0.773 | 0.379 |
| | Yes | 2,222 (21.2) | 1,737 (21.7) | 3,969 (21.4) | | |
| **Health insurance status** | | | | | | |
| | No | 1,425 (13.6) | 1,276 (15.9) | 2,701 (14.6) | 20.299 | < 0.0001 |
| | Yes | 9,077 (86.4) | 6,735 (84.1) | 15,812 (85.4) | | |
| **Marital status** | | | | | | |
| | Never married | 1,045 (9.9) | 388 (4.8) | 1,433 (7.7) | 365.775 | < 0.0001 |
| | Once | 9,283 (88.4) | 7,168 (89.5) | 16,451 (88.8) | | |
| | Two and more | 178 (1.7) | 456 (5.7) | 634 (3.4) | | |
| **Number of children** | | | | | | |
| | Has no child | 1,896 (17.8) | 823 (10.3) | 2,692 (14.5) | 266.768 | < 0.0001 |
| | One | 1,770 (16.8) | 1,131 (14.1) | 2,901 (15.7) | | |
| | Two and more | 6,867 (65.4) | 6,058 (75.6) | 12,925 (69.8) | | |
| **Exposure to violence by first-degree relatives** | | | | | | |
| | No | 9,672 (92.1) | 6,696 (83.6) | 16,368 (88.4) | 317.725 | < 0.0001 |
| | Yes | 833 (7.9) | 1,313 (16.4) | 2,146 (11.6) | | |
| **Health status** | | | | | | |
| | Excellent/good | 5,286 (50.3) | 2,782 (34.7) | 8,068 (43.6) | 530.750 | < 0.0001 |
| | Reasonable | 4,055 (38.6) | 3,658 (45.7) | 7,713 (41.7) | | |
| | Bad/very bad | 1,163 (11.1) | 1,568 (19.6) | 2,731 (14.8) | | |

*(Continued)*

**Table 1.** (Continued)

| Variables | Exposure to verbal and psychological violence | | n (%) | χ2 | P |
|---|---|---|---|---|---|
| | No | Yes | | | |
| **Husband or partner's educational level** | | | | | |
| Illiterate | 364 (3.5) | 367 (4.6) | 731 (4.0) | 180.858 | < 0.0001 |
| Elementary school | 4,184 (39.9) | 3,630 (45.3) | 7,814 (42.2) | | |
| Secondary school | 1,437 (13.7) | 1,255 (15.7) | 2,692 (14.5) | | |
| High school | 2,659 (25.3) | 1,824 (22.8) | 4,483 (24.2) | | |
| University | 1,852 (17.6) | 930 (11.6) | 2,782 (15.0) | | |
| **Husband or partner's employment status** | | | | | |
| No | 1,861 (17.7) | 1,553 (19.4) | 3,414 (18.5) | 8.368 | < 0.01 |
| Yes | 8,635 (82.3) | 6,454 (80.6) | 15,089 (81.5) | | |
| **Husband or partner's alcohol use status** | | | | | |
| No | 8,778 (83.6) | 5,893 (73.6) | 14,671 (79.3) | 277.567 | < 0.0001 |
| Yes | 1,723 (16.4) | 2,117 (26.4) | 3,840 (20.7) | | |
| **Husband or partner's gambling status** | | | | | |
| No | 10,419 (99.2) | 7,705 (96.2) | 18,124 (97.9) | 201.773 | < 0.0001 |
| Yes | 81 (0.8) | 302 (3.8) | 383 (2.1) | | |
| **Husband or partner's drug use status** | | | | | |
| No | 10,480 (99.8) | 7,934 (99.2) | 18,414 (99.6) | 46.526 | < 0.0001 |
| Yes | 16 (0.2) | 66 (0.8) | 82 (0.4) | | |
| **Husband or partner's cheating status** | | | | | |
| No | 10,109 (96.3) | 6,743 (84.3) | 16,852 (91.1) | 817.102 | < 0.0001 |
| Yes | 385 (3.7) | 1,260 (15.7) | 1,645 (8.9) | | |
| **Status of women's exposure to husband or partner's economic violence** | | | | | |
| No | 8,515 (82.8) | 4,666 (58.8) | 13,181 (72.3) | 1,293.769 | < 0.0001 |
| Yes | 1,769 (17.2) | 3,275 (41.2) | 5,044 (27.7) | | |
| **Status of exposure to husband or partner's physical violence** | | | | | |
| No | 8,934 (85.0) | 2,796 (34.9) | 11,730 (63.3) | 4,921.428 | < 0.0001 |
| Yes | 1,572 (15.0) | 5,216 (65.1) | 6,788 (36.7) | | |
| **Status of exposure to husband or partner's sexual violence** | | | | | |
| No | 10,141 (96.5) | 5,756 (71.9) | 15,897 (85.9) | 2,270.962 | < 0.0001 |
| Yes | 365 (3.5) | 2,249 (28.1) | 2,614 (14.1) | | |

When Table 2 was examined, it was observed that the variables were significant concerning the survey year, region (middle), age (25–24), educational level (elementary school, high school, university), health insurance status, marital status (never married, once), health status (reasonable, bad/very bad), number of children (two and more), and status of exposure to violence by first-degree relatives. It was observed that the variables were significant regarding the husband or partner's educational level (secondary school, high school), husband or partner's alcohol use status, husband or partner's gambling status, husband or partner's cheating status, status of exposure to husband or partner's economic violence, status of exposure to husband or partner's physical violence and status of exposure to husband or partner's sexual violence were significant.

According to the binary logistic regression model presented in Table 2, the odds of exposure to verbal and psychological violence by her husband or partner was 1.20 times higher among 2014 participants as compared to 2008. The odds of exposure to verbal and psychological violence was 1.11 times higher for those living in the Central Region compared to those

**Table 2. Estimated binary logistic regression model results and marginal effects related to factors affecting women's exposure to verbal and psychological violence.**

| Variables | OR | Std. Error | 95% CI | | Elasticity (%) | Std. Error | VIF |
|---|---|---|---|---|---|---|---|
| | | | Lower | Upper | | | |
| **Survey year (reference: 2008)** | | | | | | | |
| 2014 | 1.150[a] | 0.054 | 1.049 | 1.260 | 7.98[a] | 0.027 | 1.05 |
| **Region (reference: West)** | | | | | | | |
| South | 0.982 | 0.072 | 0.851 | 1.133 | -1.08 | 0.042 | 1.20 |
| Middle | 1.109 [c] | 0.067 | 0.986 | 1.248 | 5.92[c] | 0.034 | 1.43 |
| South | 0.890 | 0.064 | 0.772 | 1.025 | -6.89 | 0.043 | 1.30 |
| East | 1.076 | 0.069 | 0.949 | 1.221 | 4.22 | 0.037 | 1.70 |
| **Place of residence (reference: rural)** | | | | | | | |
| Urban | 0.987 | 0.050 | 0.894 | 1.089 | -0.78 | 0.029 | 1.13 |
| **Age (reference: 55+)** | | | | | | | |
| 15–24 | 1.040 | 0.129 | 0.816 | 1.325 | 2.30 | 0.073 | 3.63 |
| 25–34 | 1.212 [c] | 0.120 | 0.999 | 1.472 | 11.13[c] | 0.058 | 4.29 |
| 35–44 | 1.105 | 0.106 | 0.915 | 1.334 | 5.85 | 0.057 | 3.71 |
| 45–54 | 1.100 | 0.105 | 0.912 | 1.328 | 5.60 | 0.057 | 3.04 |
| **Educational level (reference: illiterate)** | | | | | | | |
| Elementary school | 1.152 [c] | 0.087 | 0.994 | 1.335 | 8.36[c] | 0.045 | 2.51 |
| Secondary school | 1.180 | 0.126 | 0.956 | 1.455 | 9.72 | 0.063 | 1.84 |
| High school | 1.314 [a] | 0.135 | 1.074 | 1.608 | 15.84[a] | 0.057 | 2.56 |
| University | 1.354 [b] | 0.172 | 1.055 | 1.738 | 17.50[b] | 0.073 | 2.69 |
| **Individual earning/income (reference: no)** | | | | | | | |
| Yes | 0.970 | 0.058 | 0.863 | 1.090 | -1.78 | 0.035 | 1.19 |
| **Health insurance status (reference: no)** | | | | | | | |
| Yes | 0.873 [b] | 0.059 | 0.765 | 0.997 | -7.69[b] | 0.038 | 1.06 |
| **Marital status (reference: two and more)** | | | | | | | |
| Never married | 0.624 [b] | 0.117 | 0.433 | 0.901 | -24.77[b] | 0.098 | 3.84 |
| Once | 0.576 [a] | 0.080 | 0.438 | 0.757 | -29.37[a] | 0.068 | 2.80 |
| **Health status (reference: excellent/good)** | | | | | | | |
| Reasonable | 1.292 [a] | 0.065 | 1.170 | 1.426 | 14.86[a] | 0.029 | 1.25 |
| Bad/very bad | 1.505 [a] | 0.120 | 1.288 | 1.759 | 23.22[a] | 0.044 | 1.37 |
| **Number of children (reference: has no child)** | | | | | | | |
| One child | 1.169 | 0.121 | 0.954 | 1.433 | 9.34 | 0.062 | 2.82 |
| Two and more | 1.314 [a] | 0.128 | 1.085 | 1.590 | 16.04[a] | 0.059 | 3.78 |
| **Exposure to violence by first-degree relatives (reference: no)** | | | | | | | |
| Yes | 1.788 [a] | 0.138 | 1.536 | 2.080 | 31.22[a] | 0.039 | 1.04 |
| **Husband or partner's educational level (reference: elementary school)** | | | | | | | |
| Illiterate | 0.924 | 0.117 | 0.720 | 1.185 | -4.70 | 0.076 | 1.16 |
| Secondary school | 1.196 [b] | 0.086 | 1.038 | 1.378 | 10.20[b] | 0.041 | 1.23 |
| High school | 1.176 [b] | 0.077 | 1.035 | 1.336 | 9.28[b] | 0.037 | 1.50 |
| University | 0.961 | 0.086 | 0.806 | 1.146 | -2.34 | 0.053 | 1.92 |
| **Husband or partner's employment status (reference: no)** | | | | | | | |
| Yes | 0.984 | 0.061 | 0.872 | 1.111 | -0.92 | 0.036 | 1.14 |
| **Husband or partner's alcohol use status (reference: no)** | | | | | | | |
| Yes | 1.447 [a] | 0.085 | 1.290 | 1.625 | 20.65[a] | 0.032 | 1.16 |
| **Husband or partner's gambling status (reference: no)** | | | | | | | |
| Yes | 1.583 [a] | 0.273 | 1.129 | 2.220 | 24.71[a] | 0.086 | 1.07 |
| **Husband or partner's drug use status (reference: no)** | | | | | | | |

*(Continued)*

**Table 2.** (Continued)

| Variables | | OR | Std. Error | 95% CI | | Elasticity (%) | Std. Error | VIF |
|---|---|---|---|---|---|---|---|---|
| | | | | Lower | Upper | | | |
| | Yes | 1.733 | 0.798 | 0.703 | 4.272 | 29.11 | 0.222 | 1.03 |
| **Husband or partner's cheating status (reference: no)** | | | | | | | | |
| | Yes | 2.329 [a] | 0.217 | 1.941 | 2.795 | 43.29[a] | 0.042 | 1.13 |
| **Status of exposure to husband or partner's economic violence (reference: no)** | | | | | | | | |
| | Yes | 1.870 [a] | 0.101 | 1.683 | 2.078 | 34.42[a] | 0.028 | 1.15 |
| **Status of exposure to husband or partner's physical violence (reference: no)** | | | | | | | | |
| | Yes | 1.902[a] | 0.050 | 1.803 | 2.001 | 97.20[a] | 0.024 | 1.34 |
| **Status of exposure to husband or partner's sexual violence (reference: no)** | | | | | | | | |
| | Yes | 1.369[a] | 0.087 | 1.198 | 1.539 | 64.88[a] | 0.032 | 1.26 |
| **Constant** | | -1.659 | 0.205 | -2.061 | -1.257 | | | |

[a]p < .01;

[b]p < .05;

[c]p < .10; VIF: Variance Inflation Factor

living in the Western Region. The fact that the women in the study were 25–34 years old increased odds of exposure to expected verbal and psychological violence by 1.21 times compared to women who were 55 years and older. Elementary school and high school graduate women had higher odds of exposure to verbal and psychological violence by 1.15 and 1.31 times, respectively, compared to illiterate women. According to the study it's expected that the women who had poor health were likely to have a higher chance to be expose to verbal and psychological violence by 1.51 times among the women who contributed to the study. A woman with one child had higher odds of exposure to verbal and psychological violence by 1.31 times compared to women with two and more children. Women who were exposed to violence by their first-degree relatives had higher possibility of exposure to verbal and psychological violence by 1.79 times compared to others.

A woman whose husband or partner was a secondary school graduate had a 1.20 times higher odds of exposure to verbal and psychological violence compared to a woman whose husband or partner was an elementary school graduate. A woman whose husband or partner was a high school graduate had a 1.18 times higher odds of exposure to verbal and psychological violence relative to a woman whose husband or partner was an elementary school graduate. A woman whose husband or partner used alcohol had a 1.45 times higher odds of exposure to verbal and psychological violence than others. A woman whose husband or partner was gambling had a 1.58 times higher odds of exposure to verbal and psychological violence than others. A woman whose husband or partner cheated on her had a 2.33 times higher odds of exposure to verbal and psychological violence than others. According to Table 2, a woman exposed to economic violence by her husband or partner had a higher possibility of exposure to verbal and psychological violence by 1.87 times. It was observed that a woman subjected to physical violence by her husband or partner had a 1.90 times higher odds of exposure to verbal and psychological violence. Similarly, it was witnessed that a woman exposed to sexual violence by her husband or partner had a 1.37 times higher odds of exposure to verbal and psychological violence.

According to the VIF results presented in Table 2, no variable led to multicollinearity problem between the variables. Furthermore, the marginal effects of the factors affecting women's exposure to verbal and psychological violence are presented in Table 2.

When the goodness of fit of the estimated models was examined, it was observed that the results obtained from the two models were similar.

The marginal effects of factors affecting women's exposure to verbal and psychological violence are presented in Table 2 and S1 Appendix. According to the binary logistic regression model presented in Table 2, while other variables were fixed, a woman who participated in the study in 2014 had higher possibility of exposure to verbal and psychological violence by her husband or partner by 7.98% compared to a woman who took part in the study in 2008. A woman living in the Central Region had higher possibility of exposure to verbal and psychological violence by 5.92% compared to those living in the Western Region. The fact that the women in the study were 25–34 years old increased the possibility of exposure to expected verbal and psychological violence by 11.13% compared to women who were 55 years and older. According to binary logistic regression analysis results, elementary school and high school graduate women had higher possibility of exposure to verbal and psychological violence by 8.36% and 15.84%, respectively, compared to illiterate women. A woman with health insurance had a lower possibility of exposure to verbal and psychological violence by 7.69% compared to others. An unmarried woman had a 29.37% lower possibility of exposure to verbal and psychological violence compared to a woman who was married twice or more. The fact that women who contributed to the study had bad health increased the possibility of exposure to expected verbal and psychological violence by 23.22%. A woman with one child had higher possibility of exposure to verbal and psychological violence by 16.04% compared to women with two and more children. Women who were exposed to violence by their first-degree relatives had higher possibility of exposure to verbal and psychological violence by their husbands or partners by 31.22%, compared to others.

A woman whose husband or partner was a secondary school graduate had a 10.20% higher possibility of exposure to verbal and psychological violence compared to a woman whose husband or partner was an elementary school graduate. A woman whose husband or partner was a high school graduate had a 9.28% higher possibility of exposure to verbal and psychological violence relative to a woman whose husband or partner was an elementary school graduate. A woman whose husband or partner used alcohol had a 20.65% higher possibility of exposure to verbal and psychological violence than others. A woman whose husband or partner was gambling had a 24.71% higher possibility of exposure to verbal and psychological violence than others. A woman whose husband or partner cheated on her had a 43.29% higher possibility of exposure to verbal and psychological violence than others. According to Table 2, a woman exposed to economic violence by her husband or partner had a higher possibility of exposure to verbal and psychological violence by 34.42%. It was observed that a woman subjected to physical violence by her husband or partner had a 97.20% higher possibility of exposure to verbal and psychological violence. Similarly, it was witnessed that a woman exposed to sexual violence by her husband or partner had a 64.88% higher possibility of exposure to verbal and psychological violence.

## Discussion

Violence against women is considered an important public health problem and a significant threat to human rights. Although violence is a concept that varies with time and socio-cultural structure, it has become one of the most remarked-upon issues in the world in recent years. Violence can be used not only physically but also verbally and psychologically. In fact, many studies emphasize that verbal and psychological violence is a much more serious problem than other forms of violence [16, 21, 72]. The development of policies on violence against women and the serious implementation of these policies may reduce violence against women.

Determining the factors affecting violence against women may help those implementing control policies about which issues should be given more attention in reducing and eliminating violence against women. This work determined the socio-demographic, economic, and husband- or partner-related factors affecting women's exposure to verbal and psychological violence in various regions of Turkey. Binary logistic and binary probit regression models were employed to detect these factors.

Within the scope of this study, this work aims to identify the factors that affect verbal and psychological violence against women in Turkey, an emerging country, and to determine the effectiveness of these factors. It is recognized that there are numerous studies on violence against women, primarily focusing on physical violence, and it is acknowledged that there is a need for more in-depth research on verbal and psychological violence against women, depending on various factors [6, 27, 73]. Disparate studies emphasize the need for additional studies to prevent cases for a range of reasons, including the difficulty of defining verbal and psychological violence and the failure to disclose this sort of violence owing to customs, traditions, or the desire to keep it a secret [31, 74]. The purpose of this study was to discern the main determinants of developing successful strategies to prevent exposure to verbal and psychological violence in Turkey. The micro data set obtained from National research on domestic violence against women in Turkey was used in this study. The reason for using these data is that they reflect the country in general, and this study allows international comparisons and illuminates national issues.

It was found that women who completed the survey in 2014 had a higher possibility of exposure to verbal and psychological violence compared to those from 2008. Studies in the literature have investigated this situation, with diverse results [75, 76]. Correspondingly, it would be beneficial to conduct research aimed at reducing verbal and psychological violence behaviors in the coming years by taking intensive precautions regarding the issue.

Women living in the Central Region of Turkey have a higher possibility of exposure to expected verbal and psychological violence compared to those living in the Western Region. Women residing in relatively prosperous and low-income regions are likelier to be exposed to verbal and psychological violence than women residing in other regions, even though different results might be found in similar studies [26, 31]. Even though income and welfare levels are notable causes of these regional disparities, it might be important to study these differences through in-depth research, as the literature indicates that violence is more prevalent in northern and agriculture-dominated regions [16, 77].

When the age range of women who were exposed to verbal and psychological violence was examined, the fact that they were in the age range of 25–34 increased the possibility of exposure to expected verbal and psychological violence compared to the reference group. This result corroborates the findings of previous research in the literature, and women in this age group are more prone to encounter verbal and psychological violence [18, 19, 72, 78]. Considering these results, it is evident that it would be beneficial to develop policies and take measures for the relevant age groups.

Educational level is another factor affecting women's exposure to verbal and psychological violence. In this study, the possibility of exposure to verbal and psychological violence increased as the educational level increased. Contrary to the findings of this study, other studies reported that women with lower levels of education are likelier to encounter verbal and psychological violence [79, 80]. Furthermore, this work revealed studies that could not find a significant relationship [75]. Notably, the conclusion regarding education achieved in this study may have been reached for a variety of reasons, and it would be beneficial to conduct in-depth research in other countries. In the literature, it is emphasized that there are studies with contradictory findings regarding whether an increase in the education level of women reduces

the likelihood of being exposed to violence, but it is emphasized that the expected situation is that a higher education level can reduce the likelihood of exposure to violence [81, 82].

Women's lack of health insurance increases the possibility of exposure to verbal and psychological violence. There are many studies with similar results [83–85]. Women's bad health status exacerbates the possibility of exposure to expected verbal and psychological violence. In similar studies, it was indicated that strong women with good health had less possibility of exposure to violence [83–85].

The fact that the woman had never been married or had been married once decreased the possibility of exposure to expected verbal and psychological violence. In another study, unlike this one, it was stated that unattached women aged between 15–49 had a higher possibility of exposure to psychological violence [85]. Women with one child had a higher possibility of exposure to expected verbal and psychological violence compared to women with two and more children. In another study, it was stated more children may heighten the possibility of exposure to violence [86]. Women's exposure to violence by first-degree relatives increased the possibility of exposure to verbal and psychological violence. A similar result was obtained in another study [87].

The fact that the woman's husband or partner was a secondary school graduate increased the possibility of exposure to expected verbal and psychological violence. One study found that an increase in the educational level of the husbands of women who were exposed to violence decreased the possibility of exposure to violence [79]. The research determined that the woman's husband or partner's alcohol use increased the possibility of exposure to expected verbal and psychological violence. Similar results were obtained in the studies administered in different countries [88, 89]. The fact that a woman is cheated on by her husband/partner increased the possibility of exposure to expected verbal and psychological violence. This result is consistent with that of other studies in the literature [88].

Women's exposure to economic violence raises the possibility of exposure to verbal and psychological violence. In similar studies, it has been emphasized that financial dependency heightens the likelihood of exposure to violence [51, 80]. Likewise, women who experience physical violence are likelier to be exposed to verbal and psychological violence. According to research, after exposure to partner-on-partner violence, victims experience psychological distress. Serious consequences can be encountered to the extent of suicide attempts [4, 72, 90].

Studies investigating violence against women mainly focus on physical violence. Studies on verbal violence have generally focused on healthcare professionals. The studies indicated that healthcare professionals were more exposed to verbal violence than to physical violence. Studies have shown that healthcare professionals are more frequently subjected to verbal violence than to physical violence [7, 22, 91]. Today, it is acknowledged that some instances of violence against women are kept secret and are not disclosed. In many studies, it has been determined that women and girls do not admit the instances of domestic violence they have suffered. Traditional, cultural, and psychological factors are among the causes [31, 74, 92, 93].

An individual's exposure to sexual violence also increases the possibility of exposure to verbal and psychological violence. According to the literature, women exposed to sexual violence are likelier to experience verbal and psychological violence [72, 83]. Although most studies have analyzed the prevalence and consequences of physical and sexual violence, women often think that psychological or emotional abuse may be even more harmful [94]. It is understood that this situation can lead to serious psychological consequences [29]. Studies have revealed that prenatal exposure to verbal, psychological and sexual violence has negative effects on newborns [33, 58].

Diverse forms of violence are frequently interconnected and continuous, as opposed to being isolated incidents, and form "systemic violence" [95]. It is important to recognize that

there are various forms of partner or spouse violence against women, and that there is a cause-and-effect relationship between them. Environments that nurture and witness violence will exacerbate violent behavior, and these effects will determine the direction of efforts against violence [34].

Studies have highlighted that women can adopt a wide variety of coping methods to deal with abuse, including silence, nonresponse, leaving their spouse or partner permanently or temporarily, submission, appeasement, and minimization of violence [96]. In similar studies, it has been determined that women exposed to verbal and psychological violence need training that will help them prevent and manage the violence in question, and such training can aid in the prevention of violence [7, 20].

## Conclusion

As emphasized within the scope of the study, there is a need for urgent measures to prevent this violence. More effective results can be achieved by prioritizing women in the 25–34 age group with a high possibility of exposure to violence, no health insurance, exposure to violence by first-degree relatives, exposure to physical, economic, and sexual violence, poor health status, and a husband or partner who uses alcohol in policies aimed at preventing acts of verbal violence against women in our country.

## Limitations of the study

This study has several limitations. First, the data in this study were secondary data. The variables required for statistical analysis consisted of the variables in the dataset. However, some variables, such as occupation and home ownership, that were not included in the data set could not be included in the analysis. Second, because the data are cross-sectional, the definite causal relationship between verbal violations and related socio-economic factors cannot be inferred. Third, the data on individuals' exposure to verbal and psychological violence were the individuals' own answers. Therefore, the data obtained in this data collection method may be biased. Finally, the data in the study consist of women between the ages of 15¬–59. Since a sample will be created across Turkey, women aged 60 and over were excluded from the study because the likelihood of women aged 15–59 in the houses visited was higher [60].

## Directions/suggestions for future research

There are few studies on verbal and psychological violence against women. Therefore, it will be useful to conduct relevant studies from different perspectives. Furthermore, after pandemics such as COVID-19, which caused people to lock themselves in their houses for days, the effect of the pandemic on violence against women can also be examined. In our world, where much will not be the same as it was, regional differences in verbal and psychological violence against women before and after the pandemic can be investigated.

## Supporting information

**S1 Appendix.**
(PDF)

## Acknowledgments

The authors would like to thank the Turkey Statistical Institute for its data. The views and opinions expressed in this manuscript are those of the authors only and do not necessarily represent the views, official policy, or position of the Turkey Statistical Institute.

## Author Contributions

**Conceptualization:** Ömer Alkan.

**Formal analysis:** Ömer Alkan.

**Investigation:** Ömer Alkan.

**Methodology:** Ömer Alkan.

**Resources:** Ceyhun Serçemeli, Kenan Özmen.

**Writing – original draft:** Ceyhun Serçemeli.

**Writing – review & editing:** Ömer Alkan, Ceyhun Serçemeli, Kenan Özmen.

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
