## [Decision Letter · Decision Letter 0]

10 May 2022

PONE-D-21-33583Verbal/Psychological Violence against Women in Turkey and its DeterminantsPLOS ONE

Dear Dr. Alkan,

Thank you for submitting your manuscript to PLOS ONE. After careful consideration, we feel that it has merit but does not fully meet PLOS ONE’s publication criteria as it currently stands. Therefore, we invite you to submit a revised version of the manuscript that addresses the points raised during the review process.

Your manuscript has been assessed by an expert reviewer, whose comments are appended below. As you will see, the reviewer raises several concerns regarding the contextualisation of the research within the existing literature, aspects of the methodology, and framing of the results and conclusions. Please ensure you respond to all of these points in your response to reviewers, and revise your manuscript accordingly.

We look forward to receiving your revised manuscript.

Kind regards,

Joseph Donlan

Editorial Office

PLOS ONE

Journal Requirements:

There was no funding for this study.

Reviewers' comments:

Reviewer's Responses to Questions

**Comments to the Author**

1. Is the manuscript technically sound, and do the data support the conclusions?

Reviewer #1: Partly

2. Has the statistical analysis been performed appropriately and rigorously? 

Reviewer #1: No

3. Have the authors made all data underlying the findings in their manuscript fully available?

Reviewer #1: No

4. Is the manuscript presented in an intelligible fashion and written in standard English?

Reviewer #1: Yes

5. Review Comments to the Author

Reviewer #1: This article addresses an important topic, i.e., psychological violence against women, which has received less attention than other forms of violence. Specifically, the authors use secondary data to assess correlates of psychological violence in Turkey. Despite the relevance of the topic, there are a number of issues that limit the contribution of the article to the literature. I identify some of them below and provide recommendations on how to address them.

1) Literature review: In my view, this is one the weakest sections of the article. The review is quite general and does not offer a synthesis of previous research (e.g., what do we know about the prevalence of psychological violence in Turkey and other countries, how does it compare to other forms of violence, what factors are associated with it, what are the current gaps in the literature, etc.). Because of this, the contribution of the article is unclear, as is the selection of correlates to be included in the models, which seems to be driven more by availability than theory or previous research findings. I would encourage the authors to revisit this first section to make it more specific and link it to their analytical decisions (e.g., by justifying the inclusion of variables, and stating specific hypotheses to test).

In addition to this, some assumptions are made, with statements not being backed by appropriate citations (e.g., first sentence “It is observed that violent acts are becoming increasingly widespread in today's social life”).

The following book chapter, devoted to the topic of psychological violence against women, might be helpful to incorporate in your review: Aizpurua & O´Connell (2020). Men's psychological violence against women. The SAGE Handbook of Domestic Violence. Sage.

2) Methods: I understand that the authors are using secondary data, but much information is missing from this section to properly assess the methodology of the study. At a minimum, the article should include: response rates (given their potential impact on non-response error), language(s) in which the survey was administered, data collection dates, survey mode, and any incentives offered to participants. Because two rounds/waves of data are used, information on the consistency of data collection procedures and instruments is essential to ensure that the data from 2008 and 2014 are comparable.

In addition to this, the authors should state how was missing data handled (e.g., multiple imputation, complete case analysis, in which case, it would be helpful to provide evidence that data was missing at random).

Although the operationalisation of most variables is described in this section, I don´t think the authors explain how were other forms of violence measured, and what the timeframes were. Specifying the timeframe used in the question wording is also relevant for the main outcome (psychological violence), as it will influence estimates and the ability to compare estimates from this study with prior research.

I do appreciate the authors using weights in their analysis, but I encourage them to specify what type of weights were used for transparency and replicability.

3) Results: Given the consistency of logit and probit models, it is unclear to me why the authors decided to report both. I wonder if presenting one of them and having the other models in the Appendix -and a note regarding consistency in the main text- would help streamline the findings, as the added value of having both sets of analysis in the main text is unclear at the moment.

I also have questions regarding the model strategy, and the decision to estimate one model with the women´s variables and another model with the husbands´ variables. I would, instead, suggest estimating nested models (the first one including only women-related variables, and then the full model). This way, you control for all variables in the full model and can see how model specification influences the findings. There is one article that examined correlates of psychological violence against women in Spain using secondary data which included both sets of variables, which might be helpful: Aizpurua et al. (2018). Controlling Behaviors and Intimate Partner Violence Among Women in Spain: An Examination of Individual, Partner, and Relationship Risk Factors for Physical and Psychological Abuse. Journal of interpersonal Violence. https://doi.org/10.1177/0886260517723744).

A more minor question has to do with the selection of reference categories in your models. It would be helpful to choose a consistent criterion, such as the most frequent category.

4) Discussion: Although you acknowledge the limitations of the cross-sectional research design, sometimes causal language is used, which is not appropriate. I would encourage the authors to review the article to ensure that those references are removed. I would also suggest further discussing some limitations of the research, such as the few indicators used to measure psychological violence, or the exclusion of important age groups from the population (women 60 and over). More broadly, I think this section could be streamlined by providing less details about individual studies and situating the findings within the literature a bit more broadly (what is consistent, what is inconsistent), as well as linking the finding with theory, and not only empirical findings.

I hope these comments are helpful and wish the authors the best as they move forward with this manuscript.

6. PLOS authors have the option to publish the peer review history of their article (what does this mean?). If published, this will include your full peer review and any attached files.

Reviewer #1: No

---

## [Author Response · Author response to Decision Letter 0]

10 Jul 2022

Editor’s Evaluation

Thank you for submitting your manuscript to PLOS ONE. After careful consideration, we feel that it has merit but does not fully meet PLOS ONE’s publication criteria as it currently stands. Therefore, we invite you to submit a revised version of the manuscript that addresses the points raised during the review process.

Your manuscript has been assessed by an expert reviewer, whose comments are appended below. As you will see, the reviewer raises several concerns regarding the contextualisation of the research within the existing literature, aspects of the methodology, and framing of the results and conclusions. Please ensure you respond to all of these points in your response to reviewers, and revise your manuscript accordingly.

AUTHORS’ RESPONSE

Thank you for the opportunity to revise our paper. We definitely took the feedback of the reviewers to heart and incorporated the suggested revisions.

Reviewers' comments:

Reviewer #1: This article addresses an important topic, i.e., psychological violence against women, which has received less attention than other forms of violence. Specifically, the authors use secondary data to assess correlates of psychological violence in Turkey. Despite the relevance of the topic, there are a number of issues that limit the contribution of the article to the literature. I identify some of them below and provide recommendations on how to address them.

Comment: Thank you for the comment. We definitely took the feedback of yours to heart and incorporated the suggested revisions.

1) Literature review: In my view, this is one the weakest sections of the article. The review is quite general and does not offer a synthesis of previous research (e.g., what do we know about the prevalence of psychological violence in Turkey and other countries, how does it compare to other forms of violence, what factors are associated with it, what are the current gaps in the literature, etc.). Because of this, the contribution of the article is unclear, as is the selection of correlates to be included in the models, which seems to be driven more by availability than theory or previous research findings. I would encourage the authors to revisit this first section to make it more specific and link it to their analytical decisions (e.g., by justifying the inclusion of variables, and stating specific hypotheses to test).

Comment: Thank you for the comment. The introduction and literature review of the study was re-written in detail as per the comments of the Reviewer, explaining how the study contributes to the current literature. The Turkey literature of the study was rewritten in detail.

In addition to this, some assumptions are made, with statements not being backed by appropriate citations (e.g., first sentence “It is observed that violent acts are becoming increasingly widespread in today's social life”).

The following book chapter, devoted to the topic of psychological violence against women, might be helpful to incorporate in your review: Aizpurua & O´Connell (2020). Men's psychological violence against women. The SAGE Handbook of Domestic Violence. Sage.

Comment: Thank you for the comment. The studies recommended by the Reviewer were reviewed in detail and the introduction and literature review section was rewritten to be used in the paper.

2) Methods: I understand that the authors are using secondary data, but much information is missing from this section to properly assess the methodology of the study. At a minimum, the article should include: response rates (given their potential impact on non-response error), language(s) in which the survey was administered, data collection dates, survey mode, and any incentives offered to participants. Because two rounds/waves of data are used, information on the consistency of data collection procedures and instruments is essential to ensure that the data from 2008 and 2014 are comparable.

Comment: Thank you for the comment. The methods section of the study was re-written in detail as per the comments of the Reviewer. The required explanations about the sample and data were added to the method section in accordance with the criticism of the reviewer.

In addition to this, the authors should state how was missing data handled (e.g., multiple imputation, complete case analysis, in which case, it would be helpful to provide evidence that data was missing at random).

Comment: Thank you for the comment. The required explanations about the sample and data were added to the method section in accordance with the criticism of the reviewer.

Although the operationalisation of most variables is described in this section, I don´t think the authors explain how were other forms of violence measured, and what the timeframes were. Specifying the timeframe used in the question wording is also relevant for the main outcome (psychological violence), as it will influence estimates and the ability to compare estimates from this study with prior research.

Comment: Thank you for the comment. The required correction has been performed in accordance with the criticism of the reviewer. Considering the comments of the Reviewer, “…status of exposure to husband/partner's economic violence at any point in her life (no, yes), status of exposure to husband/partner's physical violence at any point in her life (no, yes), and status of exposure to husband/partner's sexual violence at any point in her life (no, yes).” statements have been incorporated into the article under the “Measures and variables” section.

I do appreciate the authors using weights in their analysis, but I encourage them to specify what type of weights were used for transparency and replicability.

Comment: Thank you for the comment. The required correction has been performed in accordance with the criticism of the reviewer. Considering the comments of the Reviewer, “The computed weights of women were added to these data sets in accordance with the with the sample design of the research. Each cluster was assigned a different weight; the reasons for this can be summarized as follows: 1) Differential selection probabilities at the cluster level; 2) Non-proportional distribution of the sample size; and 3) Differential response rates in each stratum” statements have been incorporated into the article under the “Study size” section.

3) Results: Given the consistency of logit and probit models, it is unclear to me why the authors decided to report both. I wonder if presenting one of them and having the other models in the Appendix -and a note regarding consistency in the main text- would help streamline the findings, as the added value of having both sets of analysis in the main text is unclear at the moment.

Thank you for the comment. The result section of the study was re-written in detail as per the comments of the Reviewer.

I also have questions regarding the model strategy, and the decision to estimate one model with the women´s variables and another model with the husbands´ variables. I would, instead, suggest estimating nested models (the first one including only women-related variables, and then the full model). This way, you control for all variables in the full model and can see how model specification influences the findings. There is one article that examined correlates of psychological violence against women in Spain using secondary data which included both sets of variables, which might be helpful: Aizpurua et al. (2018). Controlling Behaviors and Intimate Partner Violence Among Women in Spain: An Examination of Individual, Partner, and Relationship Risk Factors for Physical and Psychological Abuse. Journal of interpersonal Violence. https://doi.org/10.1177/0886260517723744).

Comment: Thank you for the comment. The studies recommended by the Reviewer were reviewed in detail. We actually estimated only one model. However, we have shown the estimation results in two separate tables. This was not appropriate. The required correction has been performed in accordance with the criticism of the reviewer. We have given all the estimation results in a single table.

A more minor question has to do with the selection of reference categories in your models. It would be helpful to choose a consistent criterion, such as the most frequent category.

Comment: Thank you for the comment. Considering the comments of the Reviewer, “The problem of multicollinearity in the models was taken into account while identifying the reference category for ordinal and nominal variables having more than two categories. In this regard, the best model was tried to be estimated. Therefore, a consistent criterion could not be selected” statements have been incorporated into the article under the “Measures and variables”.

4) Discussion: Although you acknowledge the limitations of the cross-sectional research design, sometimes causal language is used, which is not appropriate. I would encourage the authors to review the article to ensure that those references are removed. I would also suggest further discussing some limitations of the research, such as the few indicators used to measure psychological violence, or the exclusion of important age groups from the population (women 60 and over). More broadly, I think this section could be streamlined by providing less details about individual studies and situating the findings within the literature a bit more broadly (what is consistent, what is inconsistent), as well as linking the finding with theory, and not only empirical findings.

Comment: Thank you for the comment. The discussion section was reorganized in accordance with the criticism of the Reviewer. Considering the comments of the Reviewer, “Finally, the data in the study consists of women between the ages of 15-59. Since a sample will be created across Turkey, women aged 60 and over were excluded from the study because the likelihood of women aged 15-59 in the visited houses was higher” statements have been incorporated into the article under the “Conclusions” section.

I hope these comments are helpful and wish the authors the best as they move forward with this manuscript.

Comment: Thank you for the comment. We definitely took the feedback of yours to heart and incorporated the suggested revisions.

---

## [Editor Report · Decision Letter 1]

13 Sep 2022

PONE-D-21-33583R1Verbal/Psychological Violence against Women in Turkey and its DeterminantsPLOS ONE

Dear Dr. Alkan,

Thank you for submitting your manuscript to PLOS ONE. After careful consideration, we feel that it has merit but does not fully meet PLOS ONE’s publication criteria as it currently stands. Therefore, we invite you to submit a revised version of the manuscript that addresses the points raised during the review process.

ACADEMIC EDITOR: 

Literature review: It is usually not expected to have a section designated as literature review in the introduction of a manuscript. However, a review of the literature would be done in the introduction section. I therefore suggest that the authors remove the subtitle "Literature review" from the introduction and further summarise the section. It is quite long.

It is important for an English language editor to review and revise the manuscript

Methodology: It should be stated that this present study was a secondary date analysis

Statistical analysis: There are some data analysis that were done and reported in the result section but was not described in the statistical analysis section. e.g VIF.  Authors should describe all the statistical analysis conducted. Please explain how the models were built.

Results:

Line 411: delete "chi-square tests"  and replace with "bivariate analysis"

Table one . change p-value = 0.0000 to p-value < 0.0001

Line 423: change "ratio" to "prevalence"

Line 426: 78.6 should be 78.6%

Line 426: Delete "great"

Line 426: please delete "great"

Table 2: please report odds ratio instead of beta. odds ratio is easily explained for logistic regression than beta. Then please interprete appropriately based on the odds ratio

Conclusion is too long. Please shorten it to the essentials

We look forward to receiving your revised manuscript.

Kind regards,

Gbenga Olorunfemi, MBBS,MSC,FMCOG,FWACS

Academic Editor

PLOS ONE
---

## [Author Response · Author response to Decision Letter 1]

19 Sep 2022

Editor’s Evaluation,

Thank you for submitting your manuscript to PLOS ONE. After careful consideration, we feel that it has merit but does not fully meet PLOS ONE’s publication criteria as it currently stands. Therefore, we invite you to submit a revised version of the manuscript that addresses the points raised during the review process.

AUTHORS’ RESPONSE

Thank you for the opportunity to revise our paper. We definitely took the feedback of the academic editor to heart and incorporated the suggested revisions.

ACADEMIC EDITOR: 

Comment: Literature review: It is usually not expected to have a section designated as literature review in the introduction of a manuscript. However, a review of the literature would be done in the introduction section. I therefore suggest that the authors remove the subtitle "Literature review" from the introduction and further summarise the section. It is quite long.

Response: Thank you for the comment. We removed the subtitle "Literature review" from the introduction. The relevant section has been shortened.

Comment: It is important for an English language editor to review and revise the manuscript.

Response: Thank you for the comment. Language and grammatical errors were corrected during the “proofreading” made by the language editor of Scribendi.

Comment: Methodology: It should be stated that this present study was a secondary date analysis

Response: Thank you for the comment. Taking this criticism into account, we added relevant expression in the subtitle Data sources/measurement.

Comment: Statistical analysis: There are some data analysis that were done and reported in the result section but was not described in the statistical analysis section. e.g VIF. Authors should describe all the statistical analysis conducted. Please explain how the models were built.

Response: Thank you for the comment. Necessary amendments were made following the comments of the academic editor.

Comment: Results:

Line 411: delete "chi-square tests" and replace with "bivariate analysis"

Table one . change p-value = 0.0000 to p-value < 0.0001

Line 423: change "ratio" to "prevalence"

Line 426: 78.6 should be 78.6%

Line 426: Delete "great"

Line 426: please delete "great"

Response: Thank you for the comment. Necessary amendments were made following the comments of the academic editor.

Comment: Table 2: please report odds ratio instead of beta. odds ratio is easily explained for logistic regression than beta. Then please interprete appropriately based on the odds ratio.

Response: Thank you for the comment. We revised Table 2. and reported odds ratio instead of beta.

Comment: Conclusion is too long. Please shorten it to the Essentials

Response: Thank you for the comment. The relevant section has been shortened. We added the subtitles Limitations of the study and Directions/suggestions for future research.

---

## [Editor Report · Decision Letter 2]

22 Sep 2022

PONE-D-21-33583R2Verbal/Psychological Violence against Women in Turkey and its DeterminantsPLOS ONE

Dear Dr. Alkan,

Thank you for submitting your manuscript to PLOS ONE. After careful consideration, we feel that it has merit but does not fully meet PLOS ONE’s publication criteria as it currently stands. Therefore, we invite you to submit a revised version of the manuscript that addresses the points raised during the review process.

We look forward to receiving your revised manuscript.

Kind regards,

Gbenga Olorunfemi, MBBS,MSC,FMCOG,FWASC

Academic Editor

PLOS ONE

Journal Requirements:

Additional Editor Comments:Introduction: Please reduce the length of the introduction. Introduction beyond 1000 words suggests a long introduction.

Italic: Please avoid italics as much as possible. You have so many italics in the manuscript

Interpretation of odds ratio. Authors should consult an experienced statistician to guide in the interpretation of odds ratio. Current odds ratio interpretations are not correct. 

---

## [Author Response · Author response to Decision Letter 2]

23 Sep 2022

Additional Editor Comments:

Comment: Introduction: Please reduce the length of the introduction. Introduction beyond 1000 words suggests a long introduction.

Response: Thank you for the comment. The introduction section has been shortened. It was shortened from 2185 words to 1045 words.

Comment: Italic: Please avoid italics as much as possible. You have so many italics in the manuscript

Response: Thank you for the comment. Necessary amendments were made following the the comments of the academic editor.

Comment: Interpretation of odds ratio. Authors should consult an experienced statistician to guide in the interpretation of odds ratio. Current odds ratio interpretations are not correct.

Response: Thank you for the comment. We consulted an experienced statistician to guide in the interpretation of odds ratio. Necessary amendments were made following the comments of the academic editor.

---

## [Editor Report · Decision Letter 3]

28 Sep 2022

Verbal/Psychological Violence against Women in Turkey and its Determinants

PONE-D-21-33583R3

Dear Dr. Alkan,

We’re pleased to inform you that your manuscript has been judged scientifically suitable for publication and will be formally accepted for publication once it meets all outstanding technical requirements.

Kind regards,

Gbenga Olorunfemi, MBBS,MSC,FMCOG,FWASC

Academic Editor

PLOS ONE
---

## [Editor Report · Acceptance letter]

29 Sep 2022

PONE-D-21-33583R3 

Verbal and Psychological Violence Against Women in Turkey and Its Determinants 

Dear Dr. Alkan:

I'm pleased to inform you that your manuscript has been deemed suitable for publication in PLOS ONE. Congratulations! Your manuscript is now with our production department. 

Kind regards, 

on behalf of

Dr. Gbenga Olorunfemi 

Academic Editor

PLOS ONE